# Real-Time Observation of Clickable Cyanotoxin Synthesis in Bloom-Forming Cyanobacteria *Microcystis aeruginosa* and *Planktothrix agardhii*

**DOI:** 10.3390/toxins16120526

**Published:** 2024-12-05

**Authors:** Rainer Kurmayer, Rubén Morón Asensio

**Affiliations:** Research Department for Limnology, University of Innsbruck, Mondseestrasse 9, 5310 Mondsee, Austria; ruben.moron-asensio@uibk.ac.at

**Keywords:** microcystin synthesis, anabaenopeptin synthesis, time-lapse experiments, molecule tracing, pulse-feeding, unnatural amino acids, click-chemistry, fluorescent labeling, growth rate

## Abstract

Recently, the use of click chemistry for localization of chemically modified cyanopeptides has been introduced, i.e., taking advantage of promiscuous adenylation (A) domains in non-ribosomal peptide synthesis (NRPS), allowing for the incorporation of clickable non-natural amino acids (non-AAs) into their peptide products. In this study, time-lapse experiments have been performed using pulsed feeding of three different non-AAs in order to observe the synthesis or decline of azide- or alkyne-modified microcystins (MCs) or anabaenopeptins (APs). The cyanobacteria *Microcystis aeruginosa* and *Planktothrix agardhii* were grown under maximum growth rate conditions (r = 0.35–0.6 and 0.2–0.4 (day^−1^), respectively) in the presence of non-AAs for 12–168 h. The decline of the azide- or alkyne-modified MC or AP was observed via pulse-feeding. In general, the increase in clickable MC/AP in peptide content reached a plateau after 24–48 h and was related to growth rate, i.e., faster-growing cells also produced more clickable MC/AP. Overall, the proportion of clickable MC/AP in the intracellular fraction correlated with the proportion observed in the dissolved fraction. Conversely, the overall linear decrease in clickable MC/AP points to a rather constant decline via dilution by growth instead of a regulated or induced release in the course of the synthesis process.

## 1. Introduction

In addition to the visually repulsive effect of cyanobacterial blooms in freshwater and brackish water, considerable amounts of toxins are produced and released into the water, with dangerous effects for the entire food web, at the end of which humans are. Within the cyanotoxins, the microcystins (MCs) and related nodularins are globally considered the most abundant [1]. To understand cyanotoxin synthesis within cyanobacterial cells, the intracellular storage mechanism of cyanotoxins, and the factors regulating cyanotoxin occurrence outside the cell are considered of outermost importance to develop exposure scenarios that are at the basis of risk management [1].

Typically, MCs are produced inside the cells while high extracellular concentrations have been observed less frequently, i.e., after cell lysis [2]. Using individual cell culture studies under laboratory conditions, both intracellular and extracellular MCs have been measured, and the general finding was that in exponentially growing cell cultures, <10% of the total MC pool is found to be extracellular. Such experiments have been performed using a few selected strains only (i.e., *Microcystis aeruginosa* PCC 7806), somewhat limiting the possibility of testing variability in ecophysiological responses via comparing a larger number of genotypes or strains [3]. For example, comparing the extracellular peptide content among 56 strains of *Planktothrix agardhii* and *P. rubescens*, the percentage of extracellular peptides (from total peptide concentration) varied from 0 to 62 ± 23% (SE) for MC and 0 to 58 ± 22% for anabaenopeptin (AP) [3,4]. While the majority of the samples (77%) were found to contain less than 10% of total MC dissolved, a few strains showed a higher proportion of dissolved MC. Nevertheless, it is generally understood that MC release into the medium is primarily related to cell aging and resulting cell lysis [5], while MC production has been traditionally linked to cell growth [6,7].

Typically, those analyses are performed in parallel to strain growth under defined culture conditions, and the intra- vs. extracellular MC or other peptide concentrations are determined by fractionation, i.e., from cells filtered on glass fiber filters, while corresponding extracellular MCs are determined from the filtrate [8]. A more direct approach to monitor MC/peptide synthesis is to create a traceable MC peptide pool and to follow the fate of both intra- and extracellular MCs during time-lapse experiments. This has been attempted in the past, i.e., by supplying radioactive inorganic carbon to *M. aeruginosa* in cultures for 6 h and subsequent incubation in the dark [5]. The fate of this radioactive MC pool was studied by following its radioactivity in time-course experiments. Notably, no evidence for losses from the intracellular MC pool could be found either under high light or under low light conditions.

In the last years, the use of click chemistry to assist in subcellular localization of chemically modified MC/APs has been applied. In particular, visualizing the synthesis of MCs and APs directly in the cell of cyanobacteria was established by using chemo-selective reaction [9], i.e., following the principle of the copper catalyzed azide-alkyne cycloaddition (CuAAC) reaction [10]. Accordingly, using chemical-analytical methods the targeted modification of MC/AP peptides through the incorporation of an alkyne or azide side chain could be shown, i.e., for MC structure in pos. 2 in *M. aeruginosa* or AP structure in exocyclic pos. 1 in *P. agardhii* [9]. The targeted peptides then become available as CuAAC reaction sites for specific fluorophores visualized using high-resolution microscopy.

The chemo-selective labeling process requires the incorporation of amino acids carrying azide or alkyne moieties into the modified MC or AP peptide to allow its targeting. The incorporation of such non-natural amino acids (non-AAs) is possible since cyanobacteria assimilate various diverse amino acids from the media [11], and certain promiscuous adenylation (A) domains integrate the non-AAs into the growing peptide product during non-ribosomal peptide synthesis (NRPS). For the first time, those promiscuous A domains were described in *M. aeruginosa* some years ago [12,13]. The encoding genes have possibly evolved in order to increase the structural diversity of the resulting bioactive peptide products, i.e., for MC in *M. aeruginosa* [13] or for AP in *P. agardhii* [14]. In particular, for MC synthesis, the first A domain encoded via the *mcy*B gene was related to the co-production of MC molecules carrying Arg, Tyr, and Leu at pos. 2 [12,13]; while for AP synthesis, the first A domain of the *apn*A gene was related to Arg, Tyr, and Lys at pos. 1 of the AP molecule [14]. However, not all genotypes have evolved towards promiscuity in relevant A domains, but rather, certain phylogenetic clades have been differentiated either by producing only AP molecules with Arg at pos. 1, or with both Arg and Tyr at pos. 1, or with three or more AA at pos. 1 (e.g., Figure 6 in Entfellner et al., 2017 [15]). Consequently, if A domains of NRPS retained their substrate specificity towards a corresponding substrate, such incorporation of non-AA into the growing peptide product will not be possible. During previous studies, two cyanobacteria have been used for click-chemistry, i.e., *M. aeruginosa* strain Hofbauer carrying natural MC variants with Tyr and Leu at pos. 2 of the MC molecule and *P. agardhii* strain no371/1 carrying Arg, Tyr, and Lys at pos. 1 of the AP molecule. While no AP synthesis could be detected for *M. aeruginosa* strain Hofbauer, *P. agardhii* strain no371/1 has lost the genetic basis for MC synthesis and is considered nontoxic [4].

The CuAAC reaction with targeted peptides has been tested for its specificity through the use of genetically modified *P. agardhii* strain NIVA-CYA126/8, for which a specific toxin or peptide such as MC or AP has been eliminated previously [14]. In particular, *P. agardhii* strain NIVA-CYA126/8 wild type as well as its peptide knock-out mutants, which have been inactivated either in MC or in cyanopeptolin or in microviridin synthesis, all showed modified clickable AP and labeling using Alexa 488 [16]. In contrast, for the NIVA-CYA126/8 mutant inactivated in AP synthesis, no labeling was observed compared with the controls [16].

In this study, we applied pulse feeding of three different non-natural L-amino acids (non-AAs), i.e., 4-azidophenylalanine (Phe-Az), N-propargyloxy-carbonyl-L-lysine (Prop-Lys), or O-propargyl-L-Tyrosine (Prop-Tyr) to track and trace the intra- and extracellular dynamics of clickable MC/AP synthesis in a time series to better understand the synthesis process in a temporal manner. Using standard peptide analysis and fragmentation via LC-MS^n^, the build up of a traceable MC/AP pool was observed. Conversely, the decline of traceable MC/AP was followed subsequent to pulse feeding and the transfer of cells to a new medium without non-AAs. The relationships between clickable MC/AP content and resulting signal intensity from subsequent labeling using Alexa Fluor488 will be reported in a forthcoming article. Most importantly, we observed a rapid increase in clickable MC/AP content by chemical-analytical analysis (maximum after 12–24 h). Conversely, after pulsed feeding, a linear decrease in the modified peptides with time was observed (0–96 h or 0–192 h). Thus, both the build up and decline point to a rather constant production of targeted MC/AP, mostly related to growth, which is in line with the current understanding of MC or other bioactive peptide synthesis in cyanobacteria.

## 2. Results

### 2.1. Growth of Cyanobacteria in Strain Cultures During Time-Lapse Experiments Using Pulsed Feeding of Non-AAs

In general both strains, *M. aeruginosa* and *P. agardhii*, grew linearly in the presence of all three non-AAs; however, one particular non-AA (Phe-Az) was frequently found to reduce growth when compared to controls (i.e., cells grown and processed under identical conditions but without substrate) or in the presence of the other two non-AAs (Prop-Lys, Prop-Tyr), as shown in Table 1 and Table 2 (Appendix A). Based on optical density (OD), the average growth rates varied from 0.58 ± 0.13 (SE) (day^−1^) during build up to 0.35 ± 0.05 during decline in the controls of *M. aeruginosa*. In contrast, during addition of Phe-Az, the growth decreased to 0.38 ± 0.15 and −0.04 ± 0.07, respectively. For *P. agardhii*, growth rates varied from 0.47 ± 0.1 to 0.22 ± 0.07 in controls during build up and decline experiments, respectively, and similarly to *M. aeruginosa*, the cell (filament) numbers as indicated by OD were found to be reduced in the presence of Phe-Az (Table 1). Analogously, based on dry weight (DW), reduced biomass was observed during Phe-Az addition, while Prop-Lys and Prop-Tyr did not affect the production of biomass (Table 2).

### 2.2. Observation of Clickable Peptide Synthesis in Cyanobacteria Strains During Time-Lapse Build Up and Decline Experiments

As reported previously, both original MC/AP structural variants, as well as new (clickable) MC/AP structural variants, were observed in the presence of all three non-AAs, i.e., demethylated MC variants DAsp-MC-YR and D-Asp-MC-LR and methylated MC variants MC-YR and MC-LR were partly transformed into modified MC, i.e., either DAsp-MC-Phe-AzR [M + H]^+^ 1056.5 and MC-Phe-AzR [M + H]^+^ 1070.5, or DAsp-MC-Prop-LysR [M + H]^+^ 1078.5 and MC-Prop-LysR [M + H]^+^ 1092.5, or DAsp-MC-Prop-TyrR [M + H]^+^ 1069.5 and MC-Prop-TyrR [M + H]^+^ 1083.5. Analogously, for AP, the original AP structural variants AP C [M + H]^+^ 809.5, AP B [M + H]^+^ 837.5, and AP-A [M + H]^+^ 844.5 were transformed into clickable AP, i.e., either AP-Phe-Az [M + H]^+^ 843.3 or AP-Prop-Lys [M + H]^+^ 891.5 or AP-Prop-Tyr [M + H]^+^ 882.5. All of these structural variants have been identified previously using MS^2^ and MS^3^ fragmentation analysis [9,16].

In general, MC/AP peptide production was maintained in the presence of all three non-AAs (Phe-Az, Prop-Lys, Prop-Tyr), i.e., using two-way repeated measures (RM)-ANOVA, the time factor was always found to be highly significant (*p* < 0.001).

Except for Phe-Az in *M. aeruginosa*, the total MC or AP contents (intracellular MC-LR equivalents in ng/mL or AP-B equivalents in ng/mL) were constantly increasing over time (Appendix A). This principal relationship between growth and peptide production resulted in total MC content ranging from 0.2 to 2.2 µg MC/mg DW during the time-lapse build up and 0.2–2.3 µg MC/mg DW during time-lapse decline. Analogously, the total AP content ranged from 0.15 to 0.7 µg AP/mg DW and 0.04–0.56 µg AP/mg DW during the time-lapse build up and time-lapse decline, respectively.

Nevertheless, when comparing MC or AP contents between non-AA treatments, significant differences in total intracellular peptide content vs. moderate differences in percentage of extracellular peptide were found (Table 3). When compared with controls, often the Phe-Az treatment changed in total intracellular peptide content, which was related to the lower growth rates (i.e., Table 1 and Table 2). In contrast, the percentage of total extracellular MC contents were found to be less altered in the presence of Prop-Tyr, Prop-Lys, or Phe-Az, while corresponding AP contents did not change significantly.

In addition, the percentages of clickable MC/AP peptides differed significantly between non-AA treatments, i.e., for MC in *M. aeruginosa*, the non-AA Prop-Tyr revealed the highest percentage of modified MC during both build up and decline experiments in both intracellular and extracellular fractions, i.e., MC-Prop-TyrR [M + H]^+^ 1069.5 and [M + H]^+^ 1083.5 followed by MC-Phe-AzR resulting in [M + H]^+^ 1056.5 and [M + H]^+^ 1070.5 and MC-Prop-LysR resulting in [M + H]^+^ 1078.5 and [M + H]^+^ 1092.5 (Table 3). In contrast, for AP in *P. agardhii*, the non-AA Prop-Lys revealed the highest percentage of modified AP, i.e., AP-Prop-Lys [M + H]^+^ 891.5 followed by AP-Phe-Az [M + H]^+^ 843.3 and AP-Prop-Tyr [M + H]^+^ 882.5 (Table 3).

Regarding the time-lapse build up, MC-Prop-TyrR showed a steep proportional increase until 24–48 h (>60% of total MC), followed by a plateau phase, but continued to increase until day 7 (168 h), resulting in > 80% of the total MC content. The less abundant MC-Phe-AzR and MC-Prop-LysR are already saturated after 24 h at >20% or >3% of total MC content, respectively (Figure 1, Appendix A). Conversely, during time-lapse decline, MC-Prop-TyrR decreased linearly from >40% down to 10% of total MC, while MC-Phe-AzR and MC-Prop-LysR showed only minor, albeit significant, decreases. Analogously to *M. aeruginosa*, AP-Prop-Lys in *P. agardhii* showed a steep increase during the first 24 h, reaching a plateau phase at 60% of the total AP content (Figure 2, Appendix A), while AP-Phe-Az and AP-Prop-Tyr saturated after 24 h at >10% or >5%, respectively. During time-lapse decline, the most abundant AP-Prop-Lys decreased linearly in proportion, while in comparison to control cells, less decrease in the percentage of µg AP-Prop-Lys per mg DW could be observed (Appendix A).

Thus, intracellular synthesis of modified MC/AP was observed, reaching a plateau rather fast (24–48 h). The efficiency of incorporation of various non-AAs into the MC/AP molecule varied between cyanobacteria strains and/or synthesis pathways. Conversely, for time-lapse decline, the proportion of modified MC/AP decreased linearly. However, clickable MC/AP peptides were removed from the cells at a slower rate than their appearance during build up.

### 2.3. Dissolved Fraction of Clickable and Natural MC/AP Peptides

In general, the ranking orders observed in the proportion of clickable MC/AP in the intracellular fraction, i.e., MC-Prop-TyrR > MC-Phe-AzR > MC-Prop-LysR in *M. aeruginosa* or AP-Prop-Lys > AP-Phe-Az > AP-Prop-Tyr in *P. agardhii* was the same as those found in the extracellular fraction (Table 3). The percentages of clickable and natural MC peptides in dissolved vs. intracellular fraction were compared for both time-lapse build up and decline experiments, and linear regression curves were calculated (Appendix A). While the slope of linear curves ranged between y = 1.52x, y = 0.52x, and y = 1.19x, the coefficient of variation of regression curves varied between R^2^ = 0.94, 0.39, 0.8 for MC-Prop-TyrR, MC-Prop-LysR, and MC-Phe-AzR, respectively. Thus, clickable MC proportion in the medium was found to be linearly related to the respective intracellular proportion and linear relationships did not differ significantly from natural MC-LR in slope (i.e., ANCOVA, interaction between grouping and covariate *p* = 0.08, 0.64, 0.86 for MC-Prop-TyrR, MC-Prop-LysR, and MC-Phe-AzR, respectively). Using the equal slope model in a one-way analysis of covariance (ANCOVA), a significant difference in intercept was found when comparing the adjusted group means of MC-Prop-TyR vs. MC-LR (*p* < 0.001).

In contrast to MC variants in *M. aeruginosa*, for clickable AP variants AP-Prop-Tyr and AP-Phe-Az in *P. agardhii*, the slope of linear regression curves was dissimilar when comparing with regression curves for natural AP-A (*p* = 0.58, *p* = 0.15 for AP-Prop-Tyr and AP-Phe-Az, respectively), i.e., AP-Prop-Tyr (R^2^ = 0.48), y = 2.41x and AP-Phe-Az (R^2^ = 0.51), y = 0.84x. For the most abundant AP-Prop-Lys, a significantly reduced slope of the regression line between intracellular vs. extracellular fraction was found (R^2^ = 0.83, y = 0.91x) when compared with natural AP-A (R^2^ = 0.85, y = 2.13x), as recorded from the same peptide extract (ANCOVA, *p* = 0.008). In other words, the proportion of AP-Prop-Lys in the medium was found to be underrepresented by a factor of 2 when compared to the proportion of natural AP-A (Appendix A). In summary, all modified MC/AP peptides could be quantified in the extracellular fraction, suggesting a more constant release procedure similar to their natural congeners.

### 2.4. Relationship of Clickable MC/AP Peptide Synthesis to Growth in Cyanobacteria Strains During Time-Lapse Experiments

Because of the variation in growth rate between non-AA treatments, it was important to find out whether growth or cell division rate could be related to clickable MC/AP net production rate. Thus, clickable MC or AP net production rates were calculated from intracellular MC-LR equivalents in ng/mL or AP-B equivalents in ng/mL by ln(x + 1) transformation and were plotted against the growth rate (day^−1^) calculated from OD (Figure 3).

Overall, the highest clickable MC/AP production rates were observed in the most rapidly growing cultures. In *M. aeruginosa*, for all non-AA treatments, a significant linear relationship between cell growth and the daily production of MC-Phe-AzR, MC-Prop-LysR and, MC-Prop-TyrR was found (Figure 3a–c). In *P. agardhii*, for AP-Phe-Az, the coefficient of determination R^2^ was lowest and no significant slope in regression was found (*p* = 0.33), while for AP-Prop-Lys and AP-Prop-Tyr, marginally or highly significant linear relationships between cell growth and the daily production were obtained (Figure 3d–f). Thus, for Phe-Az in *P. agardhii*, the synthesis of AP-Phe-Az was not coupled to growth, implying that cells stopped growing after the synthesis of AP-Phe-Az. Notably, for *M. aeruginosa*, the integration of Phe-Az into MC reduced the growth, but had less impact on the principal relationship between clickable MC synthesis rate and the (reduced) daily cellular growth. In summary, an overall influence of cell growth on clickable MC/AP peptide net production via non-AA integration was observed, implying that the principal relationship between cell growth and MC/AP production is still applicable.

### 2.5. Relationship of Clickable MC/AP Peptide Content Decline and Growth in Cyanobacteria Strains During Time-Lapse Experiments

During the time-lapse decline experiments, overall, a linear decrease in clickable (intracellular) MC/AP in proportion to total MC/AP was observed over time. Using growth rates calculated from OD (see above), the theoretical decline of modified MC/AP recorded at T0 was calculated and compared to the observed decline as reported above (Figure 1d and Figure 2d). If the decline in intracellular new MC/AP peptide is significantly faster than the theoretical decline through dilution by cellular growth, one may conclude on clickable MC/AP peptide breakdown or active vs. passive transport processes.

Two-Way Repeated Measures ANOVA was used to compare the observed and the theoretical estimates in the proportion of intracellular MC (Figure 4a–c) and AP (Figure 4d–f). The time factor was always found to be significant (*p* < 0.001) (i.e., 24–96 h for MC in *M. aeruginosa*). For MC-Phe-AzR, a much faster decline than predicted from the lowest cell division rate (Appendix A) was observed, indicating potential cell lysis. In contrast, for MC-Prop-LysR and the most abundant MC-Prop-TyrR no significant difference between theoretical and observed proportions (the grouping factor) was detected (*p* > 0.05). Thus, even for the most abundant MC-Prop-TyrR, a linear decline during time course, mostly via cell division rate or cell growth, was concluded.

Analogously, for clickable APs, the time factor showed a significant decrease (i.e., 48–196 h in *P. agardhii*). For AP-Phe-Az and AP-Prop-Tyr, the corresponding theoretical AP proportions tended to be slightly increased, which was found to be marginally significant (*p* = 0.02–0.03). For AP-Prop-Lys, the intracellular proportion appeared to decline less rapidly when compared with other clickable MC/AP peptides (Figure 4e). The slower decline could be related to the fact that AP-Prop-Lys in the medium was found to be underrepresented 2-fold when compared to the proportion of natural AP-A (Appendix A). In summary, the overall linear decrease in clickable MC/AP points to a rather constant decline via dilution by growth instead of a regulated or induced release during the synthesis process.

### 2.6. Fate of Non-Natural Amino Acids (Non-AAs) During Build Up and Decline

During all time-lapse build up and decline experiments, the quantity of the different non-AAs in the cellular and dissolved fraction was recorded in parallel to LC-MS peptide analysis. Using the same chromatographic separation conditions, the non-AAs, Phe-Az, Prop-Lys, and Prop-Tyr eluted at 5.4 min [M + H]^+^ 207.5, 3.4 min [M + H]^+^ 229.5, and 5.2 min [M + H]^+^ 220.5, respectively. Typically, during the entire build up experiments, non-AAs were detected in the cellular fractions (Appendix A). For *M. aeruginosa*, Prop-Tyr and Phe-Az decreased towards the end but remained available in cells, while Prop-Lys increased three-fold till the end of the experiment (168 h) (Appendix A). Similarly, for *P. agardhii*, Prop-Lys was increasing linearly by up to three-fold till the end of the experiment (168 h), while Prop-Tyr and Phe-Az were not detected.

Dissolved quantities of non-AA were, in general, one to two magnitudes higher than cellular contents and tended to decrease during build up experiments (Appendix A). In contrast, during signal decline, dissolved non-AAs were not detected in *M. aeruginosa*. In *P. agardhii*, dissolved non-AAs occurred at T0 but were reduced to undetectable from 48 h onwards (Appendix A). Thus, at T0, non-AAs were removed by washing through centrifugation and resuspension of the cells in BG11 medium. In contrast, during build up experiments, non-AAs remained dissolved in the medium and were available in unlimited amounts.

## 3. Discussion

This study is the first to track MC/AP synthesis by using promiscuous adenylation (A) domains in non-ribosomal peptide synthesis (NRPS), allowing for the incorporation of clickable non-AAs into the peptide products. While the results of visualization using the clickable amino acids as fluorescent tags will be reported in a forthcoming paper, the monitoring of modified MC/AP synthesis alone can lead to a better understanding of natural product synthesis in bloom-forming cyanobacteria. The following insights were obtained and will be discussed:

### 3.1. Variable Efficiency of Clickable MC/AP Production Using Different Non-AAs

In this study, a rapid increase in most clickable MC/AP peptides was observed, reaching a plateau after 24−48 h (Appendix A). Not all non-AAs were integrated into the new modified MC/AP peptide with the same efficiency. For *M. aeruginosa*, MC-Prop-TyrR was produced much more efficiently than MC-Prop-LysR or MC-Phe-AzR. For *P. agardhii*, AP-Prop-Lys was produced most efficiently, while both AP-Prop-Tyr and AP-Phe-Az were produced in lower amounts. In general, it is well-known that cyanobacteria uptake amino acids from the media [11]. Accordingly, during build up experiments, non-AAs have been detected in the cellular fraction consistently (Appendix A) and it is unlikely that non-AA intracellular availability caused the variable efficiency of integration into the peptide product.

Instead, it is known that A domains in NRPS are the key enzymes enabling the integration of unnatural AA into the growing peptide [17,18]. However, activation of non-AA must be compatible with the essential subsequent catalytic reactions, such as the step of peptide bond formation, i.e., condensation (C) domains are also known to exhibit moderate to high substrate specificity [19]. For example, an erroneously loaded non-AA by an A domain, which is not accepted by the responsible C domain, would prevent peptide bond formation, resulting in the build up of intermediate peptide products [20]. Thus, it is likely that for *M. aeruginosa* strain Hofbauer, the most efficient integration of Prop-Tyr into pos. 2 of MC-YR or MC-LR can explain its highest share in total MC pool (>80%). In a similar vein, it can be assumed that for *P. agardhii* strain no371/1, the most efficient integration of Prop-Lys into AP peptide product is likely related to the highest share of AP-Prop-Lys (>80%) in the total AP pool.

### 3.2. Effects of Non-AAs on Cell Division or Growth in Study Organisms

In this study, three different non-AAs were tested in order to better describe the influence on MC/AP production and growth. It is generally understood that non-AAs with alkyne or azide moieties react very specifically with their reaction partner, i.e., without side reactions or negative influence on native biochemical processes and growth. The specificity of the reaction is one criterion to be considered as a bio-orthogonal reaction [21]. In this study, Phe-Az addition distinctly reduced growth for *M. aeruginosa* as well as *P. agardhii* from the beginning (Table 1 and Table 2, Appendix A). It should be noted that Phe-Az (4-azido-L-phenylalanine) is one of the most versatile non-AAs in common use, i.e., it is widely applied for protein labeling through bio-orthogonal reactions [22]. On the other hand, Phe-Az has also been described as reactive via decomposition that may cause intracellular damage by reactive nitrenes, causing structural rearrangement of proteins or even crosslinking [23]. Nevertheless, despite this inherent reactivity, a significant positive relationship between MC-Phe-AzR production and cell division rate was observed for *M. aeruginosa* (Figure 3a). In contrast, as previously observed [16], *P. agardhii* cells appeared more negatively affected, filaments were shorter and looked disintegrated and only a weak relationship between AP-Phe-Az production and cell division could be observed (Figure 3d). In general, a positive relationship between the production of MC (and other toxins) and cell division rate in individual strains has been postulated by authors Orr and Jones [6], suggesting that although MCs and other toxins (such as anatoxin a) are secondary metabolites the MC net production depends primarily on the cellular growth rate. Thus, environmental conditions influence MC production rather indirectly via the cellular growth rate. Although this significant interdependence is not understood on a cellular level, it fits to the overall consistent MC production in strain cultures observed regularly under laboratory conditions, even when severe physiological stress conditions have been investigated [24]. In summary, a high degree of interdependence between non-AA containing clickable MC/AP production and cellular growth has been observed, implying that the addition of non-AAs (even if available in excess amounts) did not overrule the tight coupling between primary and secondary metabolism during maximum growth rate conditions.

### 3.3. Decrease in Targeted MC/AP Content During Time-Lapse Experiment

Overall, a rather linear decrease in the proportion of clickable MC/AP was observed over time. Cells were grown as clonal strain cultures at maximum growth rate under the specified semi-continuous culture conditions, following the turbidostatic principle [25]. Thus, the influence of physiological stress or cell lysis was considered of minor importance [26]. At the onset of this study, it was hypothesized that if the decline in clickable MC/AP labels would occur faster than expected from the theoretical decline through dilution by cellular growth, one may infer intracellular processing or active/passive release processes into the medium. This was observed for MC-Phe-AzR, implying that the decline of clickable MC occurred actively/passively under conditions when cell division was very much reduced. However, the linearity of the decrease implies a constant release process instead of a regulated or induced release in the course of the synthesis pathway (Figure 4).

On the other hand, the intracellular proportion of clickable AP-Prop-Lys appeared to decline less rapidly than observed for the MC-Prop-LysR peptides (Figure 4). Accordingly, the clickable AP-Prop-Lys peptides occurred with a lower proportion in the medium when compared to its natural congener AP-A (Appendix A). These results would imply that clickable APs were retained in the cell to some extent and not readily released into the medium as was observed for natural AP-A. Whether such a delay in the release of clickable APs into the medium can be confirmed by subcellular localization via peptide labeling remains to be proved. Preliminary results on peptide labeling using the above-mentioned chemo-selective reaction revealed an increased concentration of signal intensity near the membrane region of the cells (author’s unpublished data). If such a mechanism of intracellular compartmentalization can be confirmed via imaging analysis it would constitute an interesting observation on the intracellular enrichment of AP peptide before its release into the medium.

## 4. Conclusions

In summary, chemical-analytical methods quantitatively show the targeted synthesis of MC/AP peptides by incorporating an alkyne or azide side chain, which then enables visualization via chemo-selective reaction for linking fluorophores. The variable efficiency in clickable MC/AP production probably can be linked to the varying degree of promiscuity in accepting non-AA substrates by corresponding NRPS. The tight coupling between primary metabolism, i.e., cell division and growth with secondary metabolism, i.e., MC synthesis, can explain why clickable MC production is still related to growth even if the growth rate is significantly reduced (i.e., by Phe-Az). Conversely, the rather linear decrease in clickable MC/AP proportion in total MC/AP content with growth implies that dilution by growth is still the major responsible force in intra- and extracellular dynamics of clickable MC/AP synthesis.

## 5. Materials and Methods

### 5.1. Study Organisms and Growth Conditions

Precultures of clonal *Microcystis aeruginosa* strain Hofbauer (isolated from Neusiedlersee, Austria in 1982) and *Planktothrix agardhii* no371/1 (isolated from Moose Lake, Alberta, Canada in 2005) were maintained in a semi-continuous culture condition in BG11 medium [27] at 20 °C and photosynthetically active radiation of 50 µE m^−2^ s^−1^ (using fluorescent lamps Master TL-D 90 De Luxe 36W/965, Philips, Vienna, Austria) following the turbidostatic principle [3]. Under our culture conditions, *M. aeruginosa* strain Hofbauer grew in single cells only, while *P. agardhii* strain no371/1 continued to grow in long filaments (e.g., hundreds of micrometers in length). All experiments were performed in the same dedicated climate chamber under the specified conditions. During the experiments (i.e., years 2021–2022), the temperature ranged between 18.4 and 20.6 °C (mean 19.9 ± 0.1, *n* = 31), while irradiation varied on the shelf between 36 and 58 µE m^−2^ s^−1^ (mean 47 ± 2, *n* = 31). In general, precultures (40 mL culture volume in 250 mL transparent cell culture flasks, CytoOne T-75, Starlab, Vienna, Austria) were started from cultures maintained at maximum growth rate conditions until they reached an optical density (OD_600nm_) = 0.1 (1 cm cuvette, BioSpectrometer basic, Eppendorf AG, Vienna, Austria), then diluted to OD = 0.01 and 0.025 for *M. aeruginosa* and *P. agardhii*, respectively. As described, OD = 0.01 was equal to 662 ± 265 filaments/mL of *P. agardhii* strain no371/1 or 62,866 ± 29,059 cells/mL of *M. aeruginosa* strain Hofbauer [9]. Growth rates were calculated from OD or dry weight (DW) using the following formula: µ (day^−1^) = (lnOD_t+1_ − lnOD_t_)/Δt, where ln is the natural logarithmic function and Δt is the time in days.

The overall workflow is presented in Figure 5. For both time-lapse signal build up experiments, three technical replicates were inoculated for each time point (T0–T5) under identical conditions. At T0, treatments were supplemented once with L-isomeric non-AAs, i.e., either 4-azidophenylalanine (Phe-Az), (Carl Roth, Karlsruhe, Germany), N-propargyloxy-carbonyl-L-lysine (Prop-Lys), (Sichem, Bremen, Germany), or O-propargyl-L-Tyrosine (Prop-Tyr), (Iris Biotech GmbH, Marktredtwitz, Germany) (Figure 6), diluted in 20 mM NaOH, resulting in 0.05 mM final non-AA concentration (adding 100 µL of stock solution of 20 mM non-AA solution into 40 mL of culture volume). Control cell cultures were grown in the absence of non-AAs but supplemented with 100 µL of 20 mM NaOH only. For T0, the cells were harvested directly after adding the non-AA to the medium within 1 h. In order to observe the incorporation of the non-AA into the cyanobacterial peptides, culture flasks were harvested every 12 h for the first 48 h, with a final harvesting at 168 h.

Analogously, to observe the time-lapse decline of the peptide signal, the precultures were diluted to an OD = 0.01 and 0.05 for *M. aeruginosa* and *P. agardhii*, respectively. Three technical replicates were inoculated per strain and grown for 48 h in the presence or absence of the three non-Aas, as described above (Phe-Az, Prop-Lys, Prop-Tyr). At T0, following the 48 h incubation, cultures were centrifuged at 4000 rpm for 5 min and washed with sterile BG11 medium under sterile conditions. Cultures were centrifuged again at 4000 rpm for 5 min, resuspended in fresh non-AA free sterile BG11 medium, and transferred to fresh culture flasks. For T0, cells were harvested directly after washing and transferring into fresh BG11 medium within 1 h. Time points T1-T4 were every 24 h for *M. aeruginosa* and every 48 h for *P. agardhii*. The manipulation of cells was carried out at room temperature.

### 5.2. Cell Harvesting and Fixation

At T0–T4 (T5), cells from culture flasks (40 mL culture volume) were harvested via low-vacuum filtration using pre-weighed glass fiber filters (GF/C, Ø = 2.5 cm, ROTILABO^®^ Typ CR261, Carl Roth, Karlsruhe, Germany). The filters with the collected biomass were dried using a vacuum centrifuge at RT for 4 h and dry weight was determined. Filters were stored frozen (−20 °C) until peptide extraction.

For each sample, the extracellular MC and AP peptide concentrations were determined from GF/C filtrate using standard solid phase extraction (SPE) [28]. Briefly, C18 columns (Sep-Pak^®^ Vac tC18 cartridge, 1 cc/100 mg, 37–55 µm, Waters Corporation, Vienna, Austria) were conditioned with 3 mL of 100% methanol and equilibrated using 3 mL of BG11 medium according to the manufacturer’s instructions. Subsequently, the GF/C filtrate was allowed to flow through the column at one drop per second and the dry column was stored frozen (−20 °C) until eluting with 1 mL of 80% (*v*/*v*) methanol for HPLC-MS analysis.

### 5.3. Peptide Extraction and HPLC-MS Analysis

The intracellular peptide content was extracted from the dried biomass using aqueous methanol on ice, as described previously [3]. In brief, the filters carrying the algal biomass were dispersed with a pistil in the reaction tube and incubated in aqueous 50% (*v*/*v*) methanol (1.5 mL volume). The extracts were shaken on ice (30 min) and centrifuged, and the clear supernatant was transferred into new 2 mL reaction tubes, which were evaporated to dryness in a vacuum concentrator (no heating). This procedure was repeated two times to ensure efficient peptide extraction. After complete evaporation of the solvent (4.5 mL), the extract was stored frozen.

For Liquid Chromatography—Mass Spectrometry (LC-MS), the dried extracts were resuspended using 150 µL of 100% methanol. After complete resuspension of the pellet, 150 µL of MilliQ (MQ) water was added. Extracts were centrifuged and from the clear supernatants, 100 µL was injected into the HPLC-DAD (HP1100, Agilent, Vienna, Austria). Peptides were chromatographically separated using a LiChrospher^®^ 100, octyldecyl silane (ODS), 5 µm particle size, LiChroCART^®^ 250-4 HPLC cartridge system (Merck, Darmstadt, Germany) using a linear water/acetonitrile (0.05% trifluoroacetic acid) gradient from 80:20 to 50:50 in 45 min at a flow rate of 1 mL/min and at 30 °C oven temperature.

In order to quantify the extracellular peptide composition, the C18 columns were eluted with 1 mL of 80% (*v*/*v*) methanol, evaporated to dryness using the vacuum concentrator, and re-dissolved in 110 µL of methanol. Finally, 100 µL was injected under identical chromatographic conditions as for the intracellular peptide extract. Recovery tests using 50 ng of MC-RR [M + H]^+^ 1038.5, MC-YR [M + H]^+^ 1045.5, and MC-LR [M + H]^+^ 995.5 analytical standards (Simris Biologics, Berlin, Germany) revealed recoveries of 83 ± 3% (SE), 64 ± 11%, 73 ± 8% using UV DAD at 240 nm, respectively. For MS peak area (base peak chromatograms, BPC), the recovery results obtained in parallel were 48 ± 11%, 160 ± 36%, and 61 ± 13% for MC-RR, MC-YR, and MC-LR, respectively.

The HPLC system was coupled to an ESI-MS (Electrospray Ionization Mass Spectrometer) ion trap (amaZon SL Ion Trap MS, Bruker Daltonik, Bremen, Germany) and operated in positive ionization mode. Nitrogen and helium were used as sheath gas and collision gas, respectively (43 psi, 9 L/min, 250 °C), with 5 kV capillary voltage. MC and AP variants were assigned according to their retention time, protonated mass [M + H]^+^, and automatically recorded fragmentation patterns for MCs [29,30] or APs [31,32]. Under these conditions, the limit of detection for analytical standards MC-RR [M + H]^+^ 1038.5, MC-YR [M + H]^+^ 1045.5, and MC-LR [M + H]^+^ 995.5 was 10 ng per injection (Simris Biologics, Berlin, Germany). To compare the total MC/AP content, all peptides were quantified in equivalents of either AP-B [M + H]^+^ 837.5 or MC-LR. According to the linear regression curves y = 1 × 10^7^x + 1 × 10^8^ and y = 2 × 10^7^x − 4 × 10^8^, y was the integrated area in the base peak chromatogram (BPC), and x was ng of AP-B (1–270 ng) or MC-LR (1–180 ng) injected [9]. In order to test peptide quantification, both UV peak area (at 240 nm for MC or 210 nm for AP) and MS peak area (base peak chromatograms, BPC or extracted ion chromatograms, EIC) were compared: For MC total BPC (EIC) area vs. total UV area correlated by R^2^ = 0.90 (y = 6 × 10^−7^x) and R^2^ = 0.98 (y = 8 × 10^−7^x), signal build up, and signal decline, respectively, where y is UV_240_ peak area and x is BPC peak area. Similarly, for AP total BPC (EIC) area vs. total UV area correlated by R^2^ = 0.95 (y = 6 × 10^−7^x) and R^2^ = 0.96 (y = 4 × 10^−7^x) for signal build up and signal decline, respectively (Appendix A). To quantify MC/AP modification, the BPC (EIC) peaks were manually integrated to express the individual peak area in the percentage of the total manually integrated peak area. For LC-MS the Bruker Compass DataAnalysis version 4.2 was used.

### 5.4. Statistical Analysis

For comparing growth data and clickable MC/AP contents, the two-way repeated measures analysis of variance (two-way RM ANOVA) was used (grouping factor: three non-AA treatments and one control; time factor: T0–T4 (T5), three replicates). The time factor was always found to be significant (*p* < 0.001). Assumptions on normality (Shapiro-Wilk) or equality of variance (Brown-Forsythe) were sometimes not met (*p* < 0.05). Since ANOVA test results are generally considered to be robust if sample sizes between treatments are equal, the statistical test results were still reported. For the pairwise multiple comparison procedure, the Bonferroni *t*-test (*p* < 0.05) was used.

For comparing growth with peptide production, cellular growth rates (calculated from OD at T0) were calculated using the formula y = (ln (x)_t+1_ − ln (x)_t0_)/Δt where y is the growth rate (day^−1^), x is the OD, and Δt is the time in days (t + 1 − t0). Analogously, the clickable MC or AP net production rates were calculated from intracellular MC/AP using the following formula: clickable MC/AP production rate (day^−1^) = (ln (x + 1)_t+1_ − ln (x + 1)_t0_)/Δt, where x is either MC-LR equivalents in ng/mL or AP-B equivalents in ng/mL. The linear regression curves (Figure 3 and Figure 4) were calculated and tested according to standard linear regression analysis.

The linear regression curves between the proportion of intracellular (independent variable) vs. extracellular (dependent) MC/AP peptides were compared in terms of slope using one-way analysis of covariance (ANCOVA) via the equality of slopes model. which tested the assumption that there is no interaction between the grouping factor (i.e., modified MC/AP vs. natural MC/AP) and the covariate (intracellular proportion of corresponding MC/AP peptide). If the equal slope model was passed, the equality of the intercepts was tested by comparing the adjusted group means via pairwise multiple comparison (Holm-Sidak, *p* < 0.05). All tests were performed using Sigma plot version 14.

## Figures and Tables

**Figure 1 toxins-16-00526-f001:**
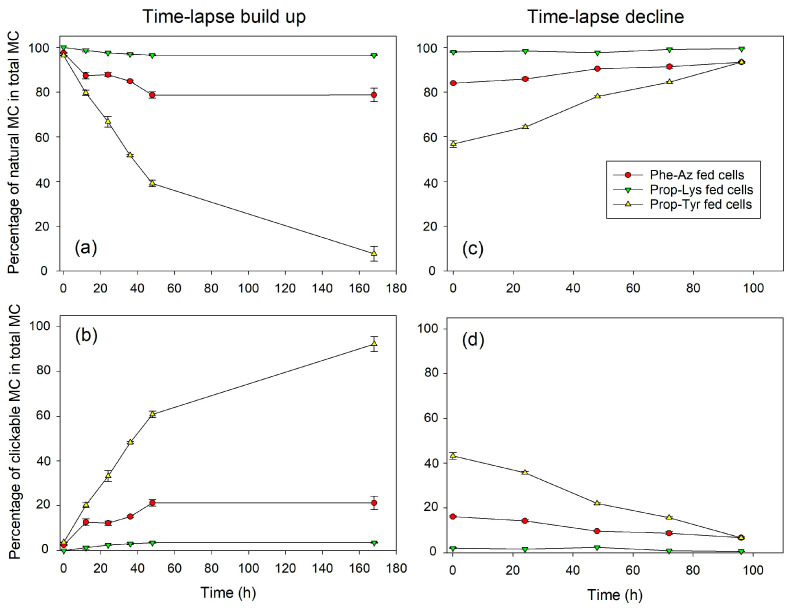
Mean (±SE) proportion of natural and clickable MC in total MC (cellular fraction, composed of four MC structural variants: DAsp-MC-YR, MC-YR, DAsp-MC-LR, MC-LR) during time-lapse experiments using pulsed feeding of non-natural amino acids (non-AAs) in order to observe the build up (**a**,**b**) or decline (**c**,**d**) of azide- or alkyne-modified MC in *M. aeruginosa* strain Hofbauer. Control cells were grown and processed under identical conditions but without non-AA substrate and could not show any clickable MC synthesis.

**Figure 2 toxins-16-00526-f002:**
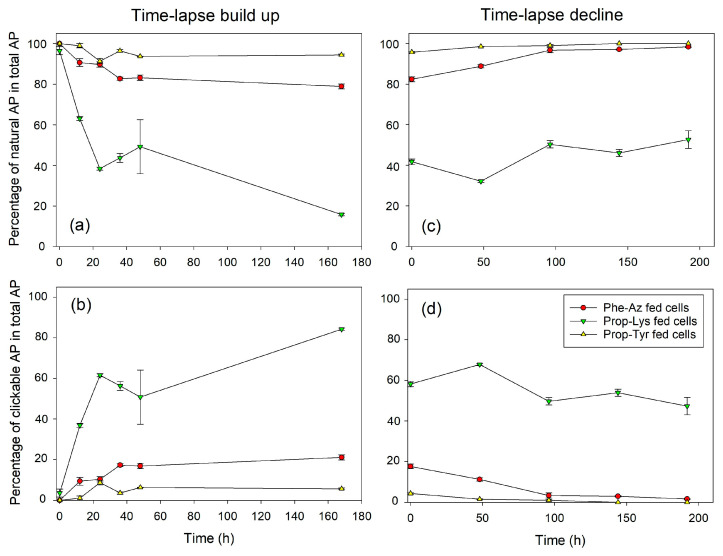
Mean (±SE) proportion of natural and clickable AP in total AP (cellular fraction, composed of four AP structural variants: unknown AP, AP-C, AP-B, AP-A) during time-lapse experiments using pulsed feeding of non-natural amino acids (non-AAs) in order to observe the build up (**a**,**b**) or decline (**c**,**d**) of azide- or alkyne modified AP in *P. agardhii* strain no371/1. Control cells were grown and processed under identical conditions but without non-AA substrate and could not show any clickable MC synthesis.

**Figure 3 toxins-16-00526-f003:**
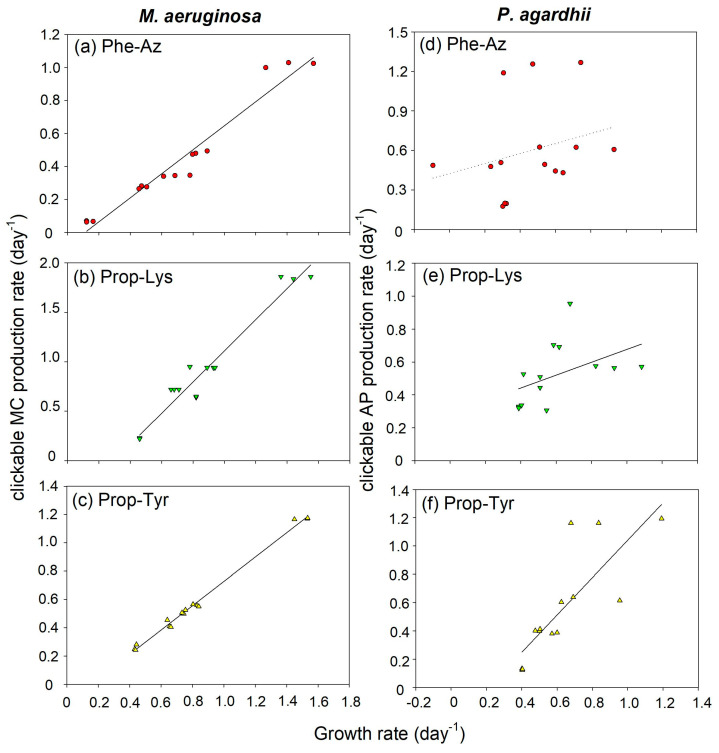
Relationship between growth rate (day^−1^) and (**a**–**c**) the clickable MC net production rate (day^−1^) in *M. aeruginosa* strain Hofbauer (calculated from ln(x + 1) MC-LR equivalents in ng/mL) and (**d**–**f**) the clickable AP net production rate (day^−1^) in *P. agardhii* strain no371/1 (calculated from ln(x + 1) AP-B equivalents in ng/mL) during time-lapse experiments using feeding of non-natural amino acids (non-AAs) in order to observe the build up of azide- or alkyne-modified MC/AP. Details of linear regression curves are as follows: (**a**) MC-Phe-AzR (y = −0.08 + 0.73x, R^2^ = 0.95, *p* < 0.0001), (**b**) MC-Prop-LysR (y = −0.47 + 1.58x, R^2^ = 0.95, *p* < 0.0001), (**c**) MC-Prop-TyrR (y = −0.13 + 0.86x, R^2^ = 0.99, *p* < 0.0001), (**d**) AP-Phe-Az (not significant, *p* = 0.33), (**e**) AP-Prop-Lys (y = 0.29 + 0.39x, R^2^ = 0.21, *p* = 0.099), (**f**) AP-Prop-Tyr (y = −0.28 + 1.32x, R^2^ = 0.65, *p* = 0.0005), where y is MC/AP production rate (day^−1^) and x is growth rate (day^−1^).

**Figure 4 toxins-16-00526-f004:**
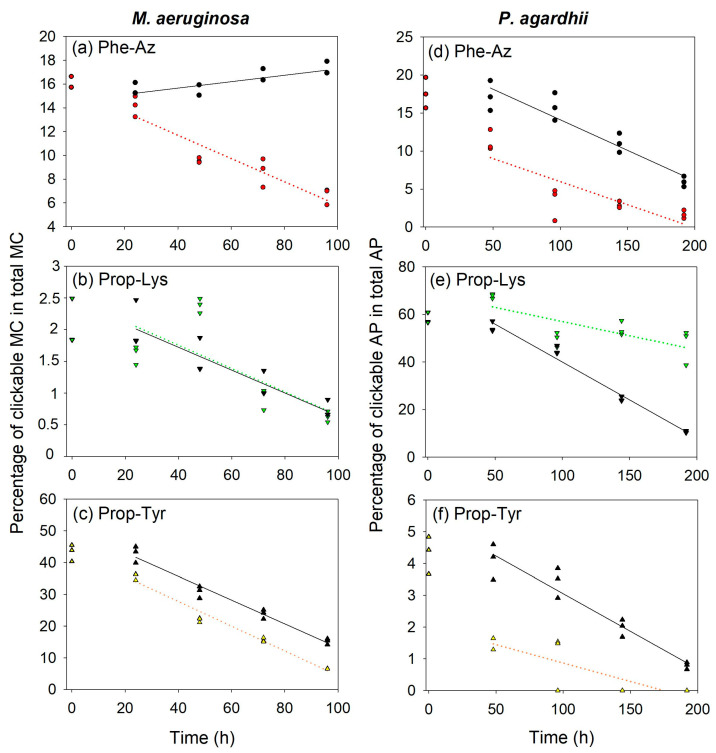
Proportion of individual clickable MC in *M. aeruginosa* (**a**–**c**) or clickable AP in *P. agardhii* (**d**–**e**) in total MC/AP (cellular fraction) during time-lapse experiments using pulsed feeding of non-natural amino acids (non-AAs) in order to observe the decline of (**a**) MC-Phe-AzR, (**b**) MC-Prop-LysR, (**c**) MC-Prop-TyrR in *M. aeruginosa,* or (**d**) AP-Phe-Az, (**e**) AP-Prop-Lys, (**f**) AP-Prop-Tyr in *P. agardhii* strain no371/1. Using growth rates, the theoretical decline of clickable MC/AP was calculated (black symbols, straight line) and compared to the observed decline (colored symbols, dotted line). Note that the scale at the *y*-axis is different, as production efficiency differs between non-AAs (Figure 1 and Figure 2).

**Figure 5 toxins-16-00526-f005:**
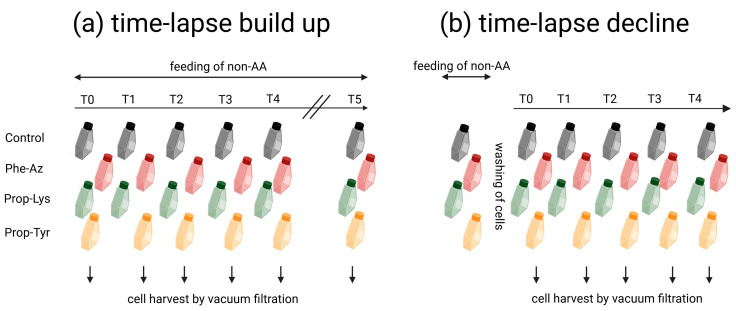
Workflow of time-lapse experiments using pulsed feeding of non-AAs for real-time observation of clickable MC/AP synthesis in bloom-forming cyanobacteria (the workflow was the same for both *M. aeruginosa* and *P. agardhii*): (**a**) time-lapse build up experiments; (**b**) time-lapse decline experiments. Created with BioRender.com. Note that the labeling of clickable peptides via chemo-selective reaction with fluorophore and high-resolution microscopy and flow-cytometry analysis using Alexa Fluor488 will be reported in a follow-up article.

**Figure 6 toxins-16-00526-f006:**
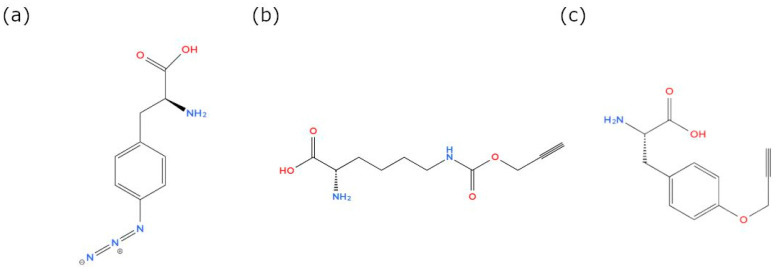
Chemical structures of non-AA molecules used for clickable microcystin (MC) synthesis in *M. aeruginosa* and for clickable anabaenopeptin (AP) synthesis in *P. agardhii*: (**a**) 4-Azido-L-phenylalanine (Phe-Az, MW 206.20 g/mol), (**b**) N-Propargyl-L-Lysine (Prop-Lys, MW 228.25 g/mol), (**c**) O-Propargyl-L-tyrosine (Prop-Tyr, MW 219.24 g/mol).

**Table 1 toxins-16-00526-t001:** Mean ± SE optical density and growth rates as recorded during time-lapse signal build up and decline experiments in cyanobacteria *M. aeruginosa* strain Hofbauer and *P. agardhii* strain no371/1. Control = cells grown and processed under identical conditions but without non-AA substrate.

Strain	Experiment	Control	Phe-Az	Prop-Lys	Prop-Tyr	*p*-Value ^1^
Optical density						
*M. aeruginosa*	build up	0.06 ± 0.02 ^a^	0.02 ± 0.002 ^b^	0.06 ± 0.02 ^a^	0.06 ± 0.02 ^a^	<0.001
	decline	0.09 ± 0.01 ^a^	0.04 ± 0.002 ^b^	0.1 ± 0.02 ^a^	0.1 ± 0.02 ^a^	<0.001
*P. agardhii*	build up	0.09 ± 0.03 ^a^	0.06 ± 0.01 ^b^	0.09 ± 0.03 ^a^	0.09 ± 0.03 ^a^	<0.001
	decline	0.34 ± 0.05 ^a^	0.29 ± 0.05 ^b^	0.37 ± 0.06 ^a^	0.36 ± 0.06 ^a^	<0.001
Growth rates (day^−1^)						
*M. aeruginosa*	build up	0.58 ± 0.13 ^ab^	0.38 ± 0.15 ^a^	0.62 ± 0.12 ^b^	0.59 ± 0.13 ^ab^	0.023
	decline	0.35 ± 0.05 ^ab^	−0.04 ± 0.07 ^b^	0.34 ± 0.06 ^ab^	0.36 ± 0.1 ^a^	0.021
*P. agardhii*	build up	0.47 ± 0.1	0.52 ± 0.27	0.53 ± 0.15	0.53 ± 0.1	0.93
	decline	0.22 ± 0.07	0.2 ± 0.04	0.2 ± 0.03	0.21 ± 0.03	0.19

^1^ Two-Way Repeated Measures ANOVA (grouping factor). Superscripts indicate homogeneous groups as revealed by post hoc pairwise comparison (Bonferroni test).

**Table 2 toxins-16-00526-t002:** Mean ± SE dry weight and growth rates as recorded during time-lapse signal build up and decline experiments in cyanobacteria *M. aeruginosa* strain Hofbauer and *P. agardhii* strain no371/1. Control = cells grown and processed under identical conditions but without non-AA substrate.

Strain	Experiment	Control	Phe-Az	Prop-Lys	Prop-Tyr	*p*-Value ^1^
Dry weight (mg/mL)						
*M. aeruginosa*	build up	0.03 ± 0.005 ^a^	0.02 ± 0.001 ^b^	0.03 ± 0.006 ^a^	0.03 ± 0.005 ^a^	0.001
	decline	0.06 ± 0.007	0.05 ± 0.004	0.05 ± 0.005	0.06 ± 0.007	0.11
*P. agardhii*	build up	0.04 ± 0.008 ^a^	0.03 ± 0.004 ^b^	0.04 ± 0.007 ^a^	0.04 ± 0.008 ^a^	<0.001
	decline	0.15 ± 0.03	0.11 ± 0.02	0.14 ± 0.02	0.16 ± 0.03	0.099
Growth rates (day^−1^)						
*M. aeruginosa*	build up	0.17 ± 0.15	0.28 ± 0.3	0.52 ± 0.3	0.21 ± 0.21	0.19
	decline	0.26 ± 0.09	0.18 ± 0.1	0.21 ± 0.13	0.28 ± 0.12	0.85
*P. agardhii*	build up	0.37 ± 0.06	0.22 ± 0.09	0.28 ± 0.17	0.31 ± 0.12	0.63
	decline	0.21 ± 0.09	0.18 ± 0.06	0.2 ± 0.05	0.24 ± 0.07	0.60

^1^ Two-Way Repeated Measures ANOVA (grouping factor). Superscripts indicate homogeneous groups as revealed by post hoc pairwise comparison (Bonferroni test).

**Table 3 toxins-16-00526-t003:** Mean ± SE total MC/AP contents (ng/mg DW), percentage extracellular peptide from total peptide and percentage of clickable MC/AP intra-and extracellular peptides recorded during time-lapse signal build up and decline experiments in cyanobacteria *M. aeruginosa* strain Hofbauer and *P. agardhii* strain no371/1. MC or AP were quantified as equivalents of MC-LR [M + H]^+^ 995.5 or AP-B [M + H]^+^ 837.5. Control = cells grown and processed under identical conditions but without non-AA substrate.

Strain	Experiment	Control	Phe-Az	Prop-Lys	Prop-Tyr	*p*-Value ^1^
Intracellular total peptide content (ng/mg DW)						
*M. aeruginosa*	build up (MC)	558 ± 49 ^a^	823 ± 120 ^b^	766 ± 117 ^ab^	677 ± 64 ^ab^	0.023
	decline (MC)	530 ± 39 ^ab^	367 ± 38 ^b^	757 ± 67 ^ab^	811 ± 119 ^a^	0.026
*P. agardhii*	build up (AP)	337 ± 39 ^ab^	267 ± 20 ^a^	299 ± 38 ^a^	418 ± 59 ^b^	0.003
	decline (AP)	258 ± 35	245 ± 25	247 ± 26	249 ± 34	0.965
Percentage of extracellular peptide from total						
*M. aeruginosa*	build up (MC)	22 ± 1.7 ^a^	21 ± 1.6 ^a^	20 ± 1.3 ^a^	26 ± 2.5 ^b^	<0.001
	decline (MC)	7.2 ± 1.2 ^a^	12.8 ± 2.2 ^b^	11.6 ± 1.4 ^ab^	10.5 ± 1.2 ^ab^	0.009
*P. agardhii*	build up (AP)	7.6 ± 0.9	6.6 ± 0.3	9.3 ± 1.6	6.2 ± 0.8	0.055
	decline (AP)	3.9 ± 0.3	5 ± 0.7	4.8 ± 0.9	4.4 ± 0.5	0.482
Percentage of clickable intracellular peptide						
*M. aeruginosa*	build up (MC)	0 ± 0 ^a^	14.1 ± 1.6 ^b^	2.2 ± 0.3 ^c^	43.2 ± 7 ^d^	<0.001
	decline (MC)	0 ± 0 ^a^	11 ± 1 ^b^	1.5 ± 0.2 ^c^	24.6 ± 3.6 ^d^	<0.001
*P. agardhii*	build up (AP)	0 ± 0 ^ad^	12.5 ± 1.7 ^ac^	50 ± 6.8 ^b^	6.4 ± 2.5 ^ac^	<0.001
	decline (AP)	0 ± 0 ^a^	7.3 ± 1.7 ^c^	55.4 ± 2.1 ^d^	1.4 ± 0.4 ^ab^	<0.001
Percentage of clickable extracellular peptide						
*M. aeruginosa*	build up (MC)	0 ± 0 ^a^	19.6 ± 3.1 ^b^	1.5 ± 0.6 ^a^	67.1 ± 9 ^c^	<0.001
	decline (MC)	0 ± 0 ^a^	10.7 ± 2.3 ^ab^	0.2 ± 0.1 ^a^	28.2 ± 7.5 ^b^	0.01
*P. agardhii*	build up (AP)	0 ± 0 ^a^	9.7 ± 2.3 ^b^	38.9 ± 6.6 ^b^	10.9 ± 3.2 ^b^	<0.001
	decline (AP)	0 ± 0 ^a^	8.7 ± 4.9 ^a^	56.2 ± 8.8 ^b^	10.4 ± 5.9 ^ab^	0.006

^1^ Two-Way Repeated Measures ANOVA (grouping factor). Superscripts indicate homogeneous groups as revealed by post hoc pairwise comparison (Bonferroni test).

## Data Availability

The original data on growth (OD, DW) and intra- vs. extracellular MC/AP peptides have been submitted to BioStudies [33] under the Accession number S-BSST1672 (DOI 10.6019/S-BSST1672) and S-BSST1681 (DOI 10.6019/S-BSST1681) (https://www.ebi.ac.uk/biostudies, accessed on 27 November 2024).

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
