# Peer review of "Real-Time Observation of Clickable Cyanotoxin Synthesis in Bloom-Forming Cyanobacteria Microcystis aeruginosa and Planktothrix agardhii"

_toxins, 2024, doi:10.3390/toxins16120526_

Round 1

Reviewer 1 Report

Comments and Suggestions for Authors

The study carries out a novel method using click chemistry to track the synthesis of cyanotoxins in two species (M. aeruginosa and P. agardhii), which provides the data in understanding the dynamics of toxin production in cyanobacteria. The manuscript is laid out well for easy catching.  The results also indicated that tracking of MC/AP synthesis by using promiscuous A domains in NRPS allowing to incorporate clickable non-AAs into peptide products. During time-lapse experiments a fast increase in clickable MC/AP content has been related to growth rate. 

specific comments:

1 Line 430, how to control the temperature and setup the irradiance? Was the greenhouse or incubator used for culturing the cells? which kind of lamp was used for offering the light?

2 L431-436, how to measure the OD? please indicate the equipment information

Author Response

Comment 1 Line 430, how to control the temperature and setup the irradiance? Was the greenhouse or incubator used for culturing the cells? which kind of lamp was used for offering the light?

Response: A dedicated climate chamber has been used. Details on temperature and light variation have been included.

Comment 2 L431-436, how to measure the OD? please indicate the equipment information.

has been added

Reviewer 2 Report

Comments and Suggestions for Authors The importance of this work lies in the fact that there is little study in the literature that focuses on the synthesis of cyanotoxins.
The manuscript is very well written and the methodology is adequately described. It therefore contains interesting and publishable data.
Just a few edits are needed to improve the quality of the manuscript before publication. The two variable amino acids at positions 2 and 4 are of L nature, the 3 non-natural amino acids used in this study, they were of what nature? It is interesting to add the chemical structures of these 3 residues in the manuscript.

Minor comment:
Page 13, 5.1. section: µEm-2S-1 instead of
µEm-1S-2  

Author Response

Comment 1: The two variable amino acids at positions 2 and 4 are of L nature, the 3 non-natural amino acids used in this study, they were of what nature? It is interesting to add the chemical structures of these 3 residues in the manuscript.

Response: the L-isomeric nature of the 3 non-natural AA has been stated. In addition the structural formulas of nonAAs have been included (Figure 6).

Comment 2: Page 13, 5.1. section: µEm-2S-1 instead of µEm-1S-2   

has been corrected